# Biotechnological Processing of Sugarcane Bagasse through Solid-State Fermentation with White Rot Fungi into Nutritionally Rich and Digestible Ruminant Feed

Nazir Ahmad Khan [1,2,*], Mussayyab Khan [2], Abubakar Sufyan [3], Ashmal Saeed [2], Lin Sun [4], Siran Wang [5], Mudasir Nazar [5], Zhiliang Tan [1], Yong Liu [1,*] and Shaoxun Tang [1,6,*]

[1] Key Laboratory for Agro-Ecological Processes in Subtropical Region, Institute of Subtropical Agriculture, Chinese Academy of Sciences, Changsha 410125, China; zltan@isa.ac.cn
[2] Department of Animal Nutrition, The University of Agriculture, Peshawar 25130, Pakistan; khanmussayyab2016@gmail.com (M.K.); ashmalsaeed123@gmail.com (A.S.)
[3] Department of Livestock and Poultry Production, Bahauddin Zakariya University, Multan 60800, Pakistan; linksufyan@bzu.edu.pk
[4] Inner Mongolia Engineering Research Center of Development and Utilization of Microbial Research in Silage, Inner Mongolia Academy of Agricultural and Animal Husbandry Sciences, Hohhot 010031, China; sunlin2013@126.com
[5] Institute of Ensiling and Processing of Grass, College of Agro-Grassland Science, Nanjing Agricultural University, Nanjing 210095, China; wangsiran@njau.edu.cn (S.W.); mudasirnazar50@yahoo.com (M.N.)
[6] College of Advanced Agricultural Sciences, University of Chinese Academy of Sciences, Beijing, 100049, China
* Correspondence: nazir.khan@aup.edu.pk (N.A.K.); y.liu@isa.ac.cn (Y.L.); sxtang@isa.ac.cn (S.T.)

**Abstract:** Sugarcane (*Saccharum officinarum*) bagasse (SCB) is one of the most widely produced lignocellulosic biomasses and has great potential to be recycled for sustainable food production as ruminant animal feed. However, due to severe lignification, i.e., lignin-(hemi)-cellulose complexes, ruminants can only ferment a minor fraction of the polysaccharides trapped in such recalcitrant lignocellulosic biomasses. This study was therefore designed to systematically evaluate the improvement in nutritional value, the in vitro dry matter digestibility (IVDMD), and the rate and extent of in vitro total gas (IVGP) and methane ($CH_4$) production during the 72 h in vitro ruminal fermentation of SCB, bioprocessed with *Agaricus bisporus*, *Pleurotus djamor*, *Calocybe indica* and *Pleurotus ostreatus* under solid-state fermentation (SSF) for 0, 21 and 56 days. The contents of neutral detergent fiber, lignin, hemicellulose and $CH_4$ production (% of IVGP) decreased ($p < 0.05$), whereas crude protein (CP), IVDMD and total IVGP increased ($p < 0.05$) after the treatment of SCB for 21 and 56 days with all white-rot fungi (WRF) species. The greatest ($p < 0.05$) improvement in CP (104.1%), IVDMD (38.8%) and IVGP (49.24%) and the greatest ($p < 0.05$) reduction in lignin (49.3%) and $CH_4$ (23.2%) fractions in total IVGP were recorded for SCB treated with *C. indica* for 56 days. Notably, *C. indica* degraded more than ($p < 0.05$) lignin and caused greater ($p < 0.05$) improvement in IVDMD than those recorded for other WRF species after 56 days. The increase in IVGP was strongly associated with lignin degradation ($R^2 = 0.72$) and a decrease in the lignin-to-cellulose ratio ($R^2 = 0.95$) during the bioprocessing of SCB. Our results demonstrated that treatment of SCB with (selective) lignin-degrading WRF can improve the nutritional value and digestibility of SCB, and *C. indica* presents excellent prospects for the rapid, selective and more extensive degradation of lignin and, as such, for the improvement in nutritional value and digestibility of SCB for ruminant nutrition.

**Keywords:** sugarcane bagasse; lignocellulosic biomass; fungal treatment; solid-state fermentation; lignin degradation; delignification; digestibility





## 1. Introduction

The efficient utilization of agriculture crop residues and the fiber-rich co-products of agro-based industries as an economical and sustainable feed or biofuel stock has become a

major research challenge in recent decades [1,2]. In ruminant nutrition, the optimal utilization of lignocellulosic biomass (LCB) could mitigate the growing food–feed competition, reduce feed costs and ensure the more complete utilization and recycling of nutrients in agriculture production systems [3,4]. Sugarcane (*Saccharum officinarum*) is among the most widely produced crops with annual productions of 2026 million tons [5], representing 21% of world's crop production. Likewise, sugarcane bagasse (SCB) is one of the most generated agro-industrial residues with a global production of 700 million tons [6], and has the potential to be recycled for sustainable food production via feeding to ruminant animals [7]. However, the utilization of SCB as ruminant feed is limited due to its low crude protein (CP) content (<3%) and large contents of highly recalcitrant lignin (>10%) and structural polysaccharides (>85%), cellulose and hemicellulose [8,9]. Ruminant animals have the ability to extract energy from cellulose and hemicellulose through symbiosis with rumen microbes. However, the lignin molecule is highly recalcitrant to microbial degradation, particularly in the low-oxygen environment of rumen. Moreover, lignin forms complexes with (hemi)cellulose, which are tightly bound together via direct (covalent) or indirect (ester or ether) linkages [2,10]. These physical and chemical complexes with lignin make the whole structure extremely dense, recalcitrant and mostly inaccessible for microbial degradation in the rumen [2,11]. Therefore, despite a very specialized digestive system, ruminants can ferment only a minor fraction of the polysaccharides embedded in the very dense recalcitrant structures of LCB such as SCB.

In recent decades, several chemical, physical, physicochemical and biotechnological pretreatment methods have been developed to degrade lignin and/or breakdown lignin–polysaccharide complexes to loosen the rigid structures of LCB and increase the accessibility of cellulose and hemicellulose for microbial fermentation in the rumen [12]. The chemical, physical and physicochemical methods are, however, expensive, require high energy input and specialized equipment and are often implicated in terms of safety issues and the production of toxic waste. In recent years, the biotechnological treatment of LCB with a wood-decaying basidiomycete white-rot fungi (WRF) in their natural growing and solid-state fermentation (SSF) conditions has emerged as a preferable treatment method [3,12]. The ecological nature, low energy requirements and greater potential for delignification compared to the other treatment methods are key features of the enormous scope of this biotechnology. This background provides a strong impetus to fully exploit this biotechnology for the optimum utilization of SCB in ruminant nutrition.

The physical structure and chemical composition of the biomass, WRF species and the SSF period are the most important factors influencing the efficiency of fungal treatment [11,13,14]. For instance, *Lentinula edodes* degraded lignin only by 4.8% in corn stover while the same fungus caused 65.4% and 65.0% delignification in rice straw and oil palm fronds, respectively [15]. This further demonstrates that the WRF species and substate combination is important for improving the nutritional value of the biomass through fungal treatment. During the biotechnological processing of biomass under SSF, the WRF not only degrade lignin but also consume cell wall polysaccharides as an energy source. Fungal species that degrade maximum lignin and minimum cellulose during the mycelial colonization stage result in maximum improvement in the nutritional value of the biomass for ruminant nutrition [16–18]. It is also evident from recent studies that the degradation of lignin during fungal treatment is strongly correlated with the rate and extent of the ruminal fermentation of the treated biomass [11,15,19]. Moreover, the extent of lignin degradation and improvement in nutritional value and fermentability of the fungal-treated biomass is positively associated with the lignin content of the untreated biomass [17]. Owing to the high lignin content (>10%) of SCB [8,9,15] compared to the commonly treated LCB, such as wheat straw (6.1–9.1%) [14], corn stover (4.95%) and rice straw (3.93%) [15], it is hypothesized that SCB has greater potential for delignification and improvement in its nutritional worth, if bioprocessed with potential WRF species. Four WRF species, i.e., *Agaricus bisporus*, *Pleurotus djamor*, *Calocybe indica* and *Pleurotus ostreatus*, were selected for this study. *Pleurotus* species were selected due to their better colonization and delignifica-

tion potential [14,20]. *C. indica* was selected as the fungus requires a low spawning rate [21] and substrate moisture level [22] and was thus expected to have greater rate and extent of substrate colonization and delignification. *A. bisporus* was selected due to its higher Mn peroxidases production potential [23], which plays a key role in lignin degradation and the early formation of mycelia [24]. Overall, the species selected for this study have been tested for treatment of cereal straws [11,25,26]. However, to our knowledge, the potential of these WRF species and treatment periods for delignification and the improvement in nutritional value and ruminal fermentation of SCB has not been evaluated.

This study was therefore designed to systematically evaluate the nutrient losses and changes in the chemical composition, the in vitro dry matter digestibility (IVDMD), and the rate and extent of in vitro total gas (IVGP) and $CH_4$ production during the 72 h in vitro ruminal fermentation of SCB, bioprocessed with *A. bisporus*, *P. djamor*, *C. indica* and *P. ostreatus* under SSF for 0, 21 and 56 days. The overall aim was to identify the most promising WRF species and SSF period in terms of improvement in the availability of nutrients in treated SCB for ruminant nutrition.

## 2. Materials and Methods

The biotechnological treatment of SCB under SSF with four WRF species was carried out at the University of Agriculture, Peshawar, Pakistan, and chemical analysis and in vitro experiments were carried out at the rumen laboratory, Department of Animal Science, Southern Illinois University, Illinois, USA. The animals used in this study were handled and cared for according to the guidelines of the ethical committee of the Southern Illinois University and the National Research Council. A brief description of the methods is given below.

### 2.1. White-Rot Fungi Species and Grain Spawn Preparation

The spores of Basidiomycete WRF, *A. bisporus*, *P. djamor*, *C. indica* and *P. ostreatus* were procured from the Department of Plant Pathology, University of Agriculture, (Peshawar, Pakistan). The spores of each fungal species were aseptically cultured on a pre-sterilized (121 °C for 20 min in autoclave) malt agar extract (pH = 5.6) plate. The agar culture contained 20.0 g/L malt extract, 0.5 g/L $KH_2PO_4$, 0.5 g/L $MgSO_4 \cdot 7H_2O$ and 0.5 g/L Ca $(NO_3)_2 \cdot 4H_2O$. After inoculation, the culture of each fungal species was incubated at 24 °C, until the mycelia had fully colonized the agar plate. The fully mycelial colonized agar plates were preserved in a refrigerator at 4 °C for grain spawn preparation.

For grain spawn production, wheat grains were first rinsed with water and then boiled in water for 20 min. The water was drained using a mesh strainer, and the moist grains were loaded into spawn mycobags (20 (w) × 60 (L) cm) until three-quarters full and subsequently autoclaved (121 °C for 60 min). The sterilized grains in mycobags were allowed to cool to room temperature (24 °C), then aseptically inoculated with 15 pieces of the mycelium-coated agar (1 $cm^2$) and sealed. The grains in mycobags were thoroughly mixed to homogenously distribute mycelium in the grains. For each fungal species, 3 separate spawn bags were prepared. All bags were then kept in an incubator (IMC18, Thermo Fisher Scientific, Waltham, MA, USA) at 24 °C until the mycelium completely colonized the grains. The prepared spawns were kept in a refrigerator at 4 °C until used for the inoculation of SCB.

### 2.2. Substrate Preparation, Pasteurization and Mycobag Processing

Sugarcane bagasse was collected from three consecutive batches (100 kg from each batch) from Khazan Sugar Mill Peshawar (Haryana Payan, Khazana, Pakistan). The bagasse was thoroughly mixed, and 50 kg representative biomass was collected and chopped to 2 cm particle size. The chopped biomass was submerged (using netted bags) in cold water for 24 h, to allow the maximum penetration of water into the biomass. The excess water was drained, and the moist bagasse was pasteurized using steam (90 °C for 2 h). After steaming, the bagasse was taken out and allowed to cool to room temperature and dry

out to a moisture content of 75%. The pasteurized bagasse (500 ± 5 g) was aseptically transferred into 64 mycobags (20 (w) × 60 (L) cm), which were further divided into four subgroups, and each subgroup (n = 16 bags per fungal species) was aseptically inoculated with grain spawn (3% on DM basis) of *A. bisporus*, *P. djamor*, *C. indica* or *P. ostreatus*. Four day-0 (control) bags, inoculated but not incubated, of each fungal species were immediately transferred to the laboratory. Subsequently, samples were collected for the analysis of dry matter (DM) content, and the remaining bagasse of each replicate day-0 bag was air dried in hot air oven at 70 °C to stop the growth of mycelium. The other inoculated bags (n = 8) of each fungal species were incubated for 21 and 56 days under the SSF conditions in a climate-controlled chamber at 24 °C with a relative humidity of 75–85%. Before the placement of the bags, 15 holes (about 0.5 cm diameter) were aseptically made in each bag for aeration and plugged with sterile cotton. At the end of each incubation period, the bags were removed and subsamples were collected for fresh DM content. The remaining bagasse of each replicate bag was air-dried in a hot air oven and processed for chemical analysis and in vitro digestibility studies.

*2.3. Chemical Analysis*

The DM content of the fresh day-0 and fungal-treated bagasse was determined by oven drying the samples at 70 °C overnight and then at 103 °C until they reached a constant weight (International Organization for Standardization (ISO), method 6496). For chemical analysis and in vitro studies, samples were air dried at 70 °C for 72 h and ground in a Thomas-Wiley Laboratory Mill (Model 4, Thomas Co., Philadelphia, PA, USA) using a 1 mm sieve. The air-dried samples were analyzed for DM content by oven drying at 103 °C until reaching a constant weight (ISO, method 6496). The CP (N × 6.25) content was analyzed with a Kjeltec 2400 autoanalyzer (Foss Analytical A/S, Hillerød, Denmark), using the Kjeldahl method (ISO, method 5983). The content of ash was determined after the complete incineration of samples in a muffle furnace at 550 °C (ISO method 5984). The ANKOM 200 Fiber Analyzer (ANKOM Technology Corps., Macedon, NY, USA) was used to analyze the contents of neutral detergent fiber (NDF) and acid detergent fiber (ADF) according to Van Soest et al. [27] methods. The lignin content was determined by the 3 h extraction of the ADF residues with 72% sulfuric acid. The NDF, ADF and lignin contents were sequentially analyzed on an ash-free basis, and the values of cellulose (ADF-lignin) and hemicellulose (NDF-ADF) were computed. The aflatoxin B1 (AFB1) rapid test kit (RingBio, Beijing, China) was used to screen the bioprocessed bagasse for AFB1 using the method described by Zhang et al. [28] for validation. The limit of detection of the kit was 5 µg/kg. The nutrient loss after 21- and 56-day treatment period with respect to the respective control was computed using the following equation:

$$Nutrient\ loss(\%) = 100 - \frac{[(100 - DM_L) \times PNP]}{PNC}$$

where $DM_L$ is the percent DM loss, PNP is the percentage of nutrient after the treatment period and PNC is the percentage of nutrient in control.

*2.4. In Vitro Dry Matter Digestibility*

One day prior to the IVDMD trial, a fermentation buffer solution was prepared in demineralized water containing 3.72 g L$^{-1}$ NaHPO$_4$, 9.82 g L$^{-1}$ NaHCO$_3$, 0.47 g L$^{-1}$ NaCl, 0.57 g L$^{-1}$ KCl, 0.15 g L$^{-1}$ CH$_4$N$_2$O, 0.12 g L$^{-1}$ MgSO$_4$·7H$_2$O and 1 mL L$^{-1}$ CaCl$_2$. The reducing solution was prepared by dissolving 320 mg NaOH and 1200 mg Na$_2$S·9H$_2$O in 200 mL of demineralized water under carbon dioxide (CO$_2$) flux. The fermentation buffer and reducing solutions were stored overnight at room temperature (24 °C) in Woulff bottles. One hour prior to the collection of rumen fluid, the buffer solution was placed in a Daisy jar assembly (SKU: D2) at 39 °C. For each run of in vitro incubation, about 500 mL of rumen fluid was obtained from each rumen-cannulated Holstein heifer (n = 4), before morning feeding in hot (39 °C) thermos bottles, pre-flushed with CO$_2$. A roughage-based total

mixed ratio containing 35% SCB, 25% alfalfa hay, 10% wheat straw and 30% concentrate mixture was fed to the animals with overall chemical composition as 7.9% ash, 12.2% CP, 1.25 Mcal/kg metabolizable energy and 52.5% neutral detergent fiber (NDF) on a DM basis. The rumen fluid was transported to the rumen laboratory, composited for four donor heifers, homogenized in a laboratory blender (LB20ES, Shanghai Prime Science Co., Ltd., Shanghai, China) for 30 s, filtered through double layers of muslin cloth and mixed (1:2 *v/v*) with the pre-warmed buffer solution in Daisy jars. The air-dried ground day-0 and pre-treated bagasse samples (1 ± 0.03 g) were placed into Dacron bags (5 cm × 10 cm) with an average pore size of 50 μm in prewarmed buffered rumen fluid in Daisy jars. The jars were flushed immediately with $CO_2$ to ensure an anaerobic environment. The reducing agent was added to the buffered rumen fluid to determine the establishment of anerobic conditions. The change in color from blue to pink and finally to almost colorless indicated the absence of oxygen. Bagasse samples from each mycobag were incubated with buffered rumen fluid in duplicate bottles in two replicate runs in the Daisy incubator (Model: D200, ANKOM Technology Corps., Macedon, NY, USA). At the end of incubation period, the Dacron bags were removed from the incubator and rinsed with deionized water until the water was transparent. After washing, all bags were dried (at 65 °C) in oven until they reached a constant weight. The IVDMD was calculated by differences in dry weight of bagasse before and after incubation. The values were corrected for blanks.

### 2.5. In Vitro Gas Production

A fully automatic wireless system of ANKOM gas production modules (Model: RFS, ANKOM Technology Corps., Macedon, NY, USA) was used to measure IVGP, $CH_4$ and $CO_2$ production, according to the procedure described by Günal et al. [29]. Briefly, samples (ca. 3 g) were weighed and placed in ANKOM bottles (SKU: 7056, 250 mL). Then, 60 mL of fresh rumen fluid and 120 mL of buffer solution (prewarmed at 39 °C) was added to each bottle and subsequently flushed with $CO_2$. Each sample was incubated in duplicate bottles with the buffered rumen fluid in two replicate runs. All bottles, including four blanks containing only rumen fluid and buffer solution, were fitted in the ANKOM modules (SKU: RF1) equipped with a pressure/gas sensor, radio frequency sender and a microchip. The bottles fitted in the modules were incubated at 39 °C in a water bath, and manually shaken for 2 min after every 2 h. The IVGP was automatically recorded by a computer with a radio frequency receiver attached, for up to 72 h of fermentation. For $CH_4$ and $CO_2$ gas production, a small aliquot of gas (10 μL) was sampled from the top of each bottle using a gas-tight syringe (Hamilton 1701 N, point style 5 needles, 51 mm; Hamilton, Bonaduz, Switzerland). The $CH_4$ and $CO_2$ contents in the gas samples were analyzed using Shimadzu GC-2030 gas chromatograph system (Shimadzu Scientific Instruments AOC-20i Plus, Columbia, MD, USA), equipped with a 100 m Rt-2560 column (0.32 mm × 0.20 μm column; Restek, Bellefonte, PA, USA). The results of IVGP were corrected for blanks in which only buffer solution and rumen liquor were incubated without feed samples.

### 2.6. Statistical Analysis

Data on the effects of fungal species, treatment period and interactions with the changes in chemical composition, $DM_L$, nutrient losses, IVDMD, IVGP, $CH_4$, $CO_2$ and AFB1 were analyzed using the PROC MIXED procedure at SAS (SAS Inst., Inc., Cary, NC, USA). The fungal species, treatment period and their interactions were fixed effects, and replications were considered random effects.

$$Y_{ijk} = \mu + FS_i + PP_j + FS_i \times PP_j + \omega_{ijk}$$

where $Y_{ijk}$ is the observation $j$ in treatment $i$; $\mu$ is the overall mean; $FS_i$ is the fixed effect of fungal species $i$ (*A. bisporus*, *P. djamor*, *C. indica* and *P. ostreatus*); $PP_j$ is the fixed effect of treatment period $j$ (0, 21 and 56 days); $FS_i \times PP_j$ is the fixed effect of interaction between fungal species $i$ and treatment period $j$; and $\omega_{ijk}$ is the random error. For parameters which showed an overall significant effect ($p < 0.05$), post hoc analyses were carried out on the

least squares means adjusted for multiple comparisons using the Tukey–Kramer test to determine significant differences between the means.

## 3. Results

### 3.1. Changes in Chemical Composition of Sugarcane Bagasse Bioprocessed with WRF Species for Different Treatment Periods

Table 1 summarizes the data on the effect of WRF species and treatment periods on the changes in the chemical composition of SCB. Except for DM, ADF and cellulose, a significant ($p < 0.01$) interaction effect of fungal species and treatment period was observed for the contents of all measured chemical components. The contents of ash and CP increased ($p < 0.05$), while the contents of NDF, lignin and hemicellulose decreased ($p < 0.05$) with the SCB treatment for 21 and 56 days for all WRF species. The further comparison of the interaction data revealed that the greatest increase ($p < 0.05$) in CP content and the greatest decrease ($p < 0.05$) in NDF and lignin contents were observed for the treatment of SCB with *C. indica* for 56 days. Notably, SCB treated with *C. indica* for 21 days had lower ($p < 0.05$) lignin content (8.40%) than the value (9.30%) recorded for *A. bisporus* after 56 days, whereas the lignin contents SCB treated with *P. djamor* (8.33%) and *P. ostreatus* (8.27%) for 56 days were comparable ($p > 0.05$) to those of *C. indica* after 21 days. On the other hand, the CP content in SCB after 21 days of treatment with *C. indica* was comparable ($p > 0.05$) to the CP content recorded after 56 days of treatment with SCB for *A. bisporus*, *P. djamor* and *P. ostreatus*.

**Table 1.** Changes in chemical composition of sugarcane bagasse bioprocessed with four white-rot fungi (WRF) species under solid-state fermentation for 0, 21 and 56 days.

| Treatment Period (Days) | WRF Species | DM (% FM) | Concentration (% DM) | | | | | | |
|---|---|---|---|---|---|---|---|---|---|
| | | | Ash | CP | NDF | ADF | Lignin | HC | CEL |
| 0 | *A. bisporus* | 24.9 | 5.09 [a] | 1.99 [a] | 86.3 [g] | 58.5 | 12.9 [e] | 27.8 [c] | 46.5 |
| | *P. djamor* | 25.4 | 5.04 [a] | 2.02 [a] | 85.1 [g] | 57.3 | 12.0 [e] | 27.8 [c] | 45.3 |
| | *C. indica* | 25.8 | 5.08 [a] | 2.08 [a] | 85.6 [g] | 57.8 | 12.1 [e] | 27.8 [c] | 45.8 |
| | *P. ostreatus* | 25.2 | 5.09 [a] | 1.98 [a] | 86.5 [g] | 58.4 | 12.0 [e] | 28.0 [c] | 46.4 |
| 21 | *A. bisporus* | 23.2 | 6.23 [b] | 2.86 [b] | 79.4 [f] | 53.8 | 10.5 [d] | 25.6 [bc] | 43.3 |
| | *P. djamor* | 24.6 | 6.27 [b] | 3.60 [c] | 73.7 [d] | 52.2 | 10.0 [cd] | 21.5 [a] | 42.2 |
| | *C. indica* | 22.9 | 6.34 [b] | 4.17 [d] | 72.0 [c] | 50.7 | 8.40 [b] | 21.6 [a] | 42.3 |
| | *P. ostreatus* | 22.9 | 6.73 [bc] | 3.77 [c] | 73.9 [d] | 53.0 | 10.1 [cd] | 20.9 [a] | 42.8 |
| 56 | *A. bisporus* | 22.8 | 7.11 [cd] | 3.98 [d] | 75.6 [e] | 50.7 | 9.30 [c] | 24.9 [b] | 41.4 |
| | *P. djamor* | 24.1 | 7.75 [e] | 4.24 [d] | 69.1 [b] | 49.0 | 8.33 [b] | 20.1 [a] | 40.7 |
| | *C. indica* | 21.8 | 7.45 [de] | 5.02 [e] | 67.2 [a] | 47.5 | 7.23 [a] | 19.7 [a] | 40.3 |
| | *P. ostreatus* | 21.4 | 7.39 [de] | 4.30 [d] | 69.6 [b] | 50.4 | 8.27 [b] | 19.2 [a] | 42.1 |
| SEM | | 0.48 | 0.19 | 0.077 | 0.85 | 0.53 | 0.17 | 0.51 | 0.50 |
| Overall mean fungal species | | | | | | | | | |
| *A. bisporus* | | 23.6 [a] | 6.14 | 2.94 [a] | 80.4 [a] | 54.3 [c] | 10.6 [c] | 26.1 [b] | 43.7 [b] |
| *P. djamor* | | 24.7 [b] | 6.35 | 3.28 [b] | 76.0 [c] | 52.8 [ab] | 10.1 [b] | 23.1 [a] | 42.7 [a] |
| *C. indica* | | 23.5 [a] | 6.29 | 3.76 [c] | 75.1 [d] | 52.0 [a] | 9.20 [a] | 23.0 [a] | 42.8 [a] |
| *P. ostreatus* | | 23.2 [a] | 6.40 | 3.35 [b] | 76.9 [b] | 53.9 [bc] | 10.1 [b] | 22.7 [a] | 43.8 [b] |
| SEM | | 0.38 | 0.16 | 0.044 | 0.41 | 0.31 | 0.09 | 0.30 | 0.29 |
| Overall mean treatment period | | | | | | | | | |
| 0 days | | 25.3 [c] | 5.07 [a] | 2.02 [a] | 85.9 [c] | 58.0 [c] | 12.0 [c] | 27.9 [c] | 46.0 [c] |
| 21 days | | 23.4 [b] | 6.39 [b] | 3.60 [b] | 74.8 [b] | 52.4 [b] | 9.77 [b] | 22.4 [b] | 42.7 [b] |
| 56 days | | 22.5 [a] | 7.43 [c] | 4.39 [c] | 70.4 [a] | 49.4 [a] | 8.28 [a] | 21.0 [a] | 41.1 [a] |
| SEM | | 0.38 | 0.052 | 0.038 | 0.13 | 0.26 | 0.083 | 0.26 | 0.25 |
| Significance | | | | | | | | | |
| WRF species | | *** | NS | *** | *** | *** | *** | *** | * |
| Treatment periods | | ** | *** | *** | *** | *** | *** | *** | *** |
| WRF species × incubation time | | NS | ** | *** | *** | NS | *** | *** | NS |

Values with different superscripts in columns for treatment periods × fungal species, overall means of fungal species or overall means of treatment periods are significantly ($p < 0.05$) different. DM, dry matter; FM, fresh matter; CP, crude protein; NDF, neutral detergent fiber; ADF, acid detergent fiber; HC, hemicellulose; CEL, cellulose; *A. bisporus*, *Agaricus bisporus*; *P. ostreatus*, *Pleurotus ostreatus*; *C. indica*, *Calocybe indica*; *P. djamor*, *Pleurotus djamor*; SEM, standard error of the mean; NS, non-significant ($p > 0.05$); *, $p < 0.05$; **, $p < 0.01$; ***, $p < 0.001$.

Among the fungal species, on average, the highest ($p < 0.05$) content of CP and lowest ($p < 0.05$) contents of NDF and lignin were recorded for *C. indica*, whereas the lowest ($p < 0.05$) content of CP and highest ($p < 0.05$) contents of NDF, ADF, lignin and hemicellulose were recorded for *A. bisporus*. The mean content of ash and CP consistently increased ($p < 0.05$) with increases in treatment period (0 to 56 days), whereas the mean contents of NDF, ADF, lignin, cellulose and hemicellulose consistently decreased ($p < 0.05$) with increases in the treatment period.

### 3.2. Dry Matter and Nutrient Losses during Bioprocessing of Sugarcane Bagasse with Four WRF Species

Data on DM and nutrient losses after 21 and 56 days of the bioprocessing of SCB under SSF with the four WRF species, in comparison to the respective controls, are summarized in Table 2. Significant ($p < 0.01$) interaction effects of fungal species and treatment periods were observed for the losses of DM and all measured chemical components. The losses (degradation) of lignin increased ($p < 0.05$) with increases in treatment period from 21 to 56 days, irrespective of the fungal species, whereas the ash and CP content improved ($p < 0.05$) with increases in treatment period. The negative values of CP and ash indicated a relative increase in their contents (Table 2). The maximum ($p < 0.05$) improvement in CP (104.1%) and maximum losses of lignin (49.3%), NDF (33.6%) and cellulose (25.6%) were recorded for SCB treated with *C. indica* for 56 days. Further comparisons show that *C. indica* caused more ($p < 0.05$) losses of lignin (38.2%) during 21 days of treatment than those caused by *A. bisporus* (28.8%) and *P. djamor* (34.2%) after 56 days.

**Table 2.** Dry matter (DM) and nutrient losses during bioprocessing of sugarcane bagasse with four white-rot fungi (WRF) species under solid-state fermentation conditions for 21 and 45 days, in comparison to day-0 (inoculated but not incubated) bagasse.

| Treatment Period (Days) | WRF Species | Percentage Losses of Nutrients in Comparison to Day 0 | | | | | | | |
|---|---|---|---|---|---|---|---|---|---|
| | | DM | Ash | CP | NDF | ADF | Lignin | HC | CEL |
| 21 | *A. bisporus* | 6.69 [abc] | −14.3 [cd] | −33.0 [d] | 14.2 [a] | 14.1 [ab] | 18.1 [a] | 14.2 [a] | 13.1 [ab] |
| | *P. djamor* | 3.27 [a] | −20.5 [bcd] | −72.4 [c] | 16.2 [b] | 11.9 [a] | 19.0 [a] | 25.0 [bc] | 10.0 [a] |
| | *C. indica* | 11.2 [d] | −10.9 [d] | −77.9 [bc] | 25.0 [e] | 22.2 [cd] | 38.2 [e] | 30.8 [c] | 17.9 [bc] |
| | *P. ostreatus* | 9.11 [cd] | −20.2 [bcd] | −72.9 [bc] | 22.3 [d] | 17.6 [bc] | 22.9 [b] | 32.1 [cd] | 16.1 [ab] |
| 56 | *A. bisporus* | 8.4 [bcd] | −27.9 [b] | −83.6 [bc] | 19.7 [c] | 20.5 [c] | 28.8 [c] | 18.1 [ab] | 18.4 [bc] |
| | *P. djamor* | 5.4 [ab] | −45.7 [a] | −98.9 [bc] | 23.2 [de] | 19.1 [bc] | 34.2 [d] | 31.7 [cd] | 15.1 [ab] |
| | *C. indica* | 15.5 [e] | −24.0 [bc] | −104.1 [a] | 33.6 [g] | 30.5 [e] | 49.3 [f] | 40.0 [de] | 25.6 [d] |
| | *P. ostreatus* | 15.1 [e] | −23.3 [bcd] | −84.3 [b] | 31.7 [f] | 26.5 [d] | 41.5 [e] | 41.8 [e] | 23.0 [cd] |
| SEM | | 0.74 | 2.58 | 2.41 | 0.39 | 1.06 | 0.71 | 1.67 | 1.03 |
| Overall mean fungal species | | | | | | | | | |
| *A. bisporus* | | 7.56 [b] | −21.1 [b] | −58.2 [c] | 17.0 [a] | 17.3 [a] | 23.5 [a] | 16.1 [a] | 15.8 [a] |
| *P. djamor* | | 4.31 [a] | −33.1 [a] | −85.6 [a] | 19.8 [b] | 15.5 [a] | 26.6 [b] | 28.3 [b] | 12.5 [a] |
| *C. indica* | | 13.4 [c] | −17.4 [c] | −90.9 [a] | 29.3 [d] | 26.3 [c] | 43.8 [d] | 35.4 [c] | 21.7 [b] |
| *P. ostreatus* | | 12.1 [c] | −21.7 [b] | −78.6 [b] | 27.0 [c] | 22.2 [b] | 32.2 [c] | 36.9 [c] | 19.8 [b] |
| SEM | | 0.52 | 1.83 | 1.70 | 0.28 | 0.78 | 0.50 | 1.23 | 0.89 |
| Overall mean treatment period | | | | | | | | | |
| 21 days | | 7.58 [a] | −16.5 [b] | −64.3 [b] | 19.4 [a] | 16.4 [a] | 24.6 [a] | 25.5 [a] | 21.0 [a] |
| 56 days | | 11.1 [b] | −30.2 [a] | −92.7 [a] | 27.1 [b] | 24.2 [b] | 38.5 [b] | 32.9 [b] | 49.1 [b] |
| SEM | | 0.37 | 1.29 | 1.20 | 0.21 | 0.75 | 0.70 | 0.87 | 0.63 |
| Significance | | | | | | | | | |
| WRF species | | *** | *** | *** | *** | *** | *** | *** | *** |
| Treatment periods | | *** | *** | *** | *** | *** | *** | *** | *** |
| WRF species × treatment period | | * | ** | *** | *** | ** | *** | ** | ** |

Values with different superscripts in columns for treatment periods × fungal species, overall means of fungal species or overall mean of treatment periods are significantly ($p < 0.05$) different. DM, dry matter; CP, crude protein; NDF, neutral detergent fiber; ADF, acid detergent fiber; HC, hemicellulose; CEL, cellulose; *A. bisporus*, *Agaricus bisporus*; *P. ostreatus*, *Pleurotus ostreatus*; *C. indica*, *Calocybe indica*; *P. djamor*, *Pleurotus djamor*; SEM, standard error of the mean; *, $p < 0.05$; **, $p < 0.01$; ***, $p < 0.001$.

The overall mean of fungal species revealed maximum ($p < 0.05$) improvement in CP content (90.9%) and maximum losses ($p < 0.05$) of lignin (43.8%), NDF (29.3%) and ADF (26.3%) for *C. indica*. The overall losses of DM, NDF, ADF, lignin, cellulose and

hemicellulose increased ($p < 0.05$) with the increase in treatment period from 21 to 56 days. In contrast, ash and CP contents improved ($p < 0.05$) with increases in treatment period.

### 3.3. Effect of Bioprocessing of Sugarcane Bagasse with Different WRF Species on In Vitro Dry Matter Digestibility, In Vitro Total Gas and $CH_4$ Production

Data on the changes in IVDMD, IVGP and $CH_4$ production of SCB bioprocessed with the four WRF species under SSF for 0, 21 and 56 days are summarized in Table 3. Except for $CH_4$ (mL/g organic matter (OM)), significant ($p < 0.05$) interaction effects of fungal species and treatment period were observed for all measured in vitro fermentation parameters (Table 3). The IVDMD and total IVGP increased ($p < 0.05$) with the treatment of SCB for 21 and 56 days with all WRF species.

**Table 3.** Changes in in vitro dry matter degradation (IVDMD), in vitro gas production (IVGP), methane ($CH_4$) emission and aflatoxin B1 (AFB1) content sugarcane bagasse bioprocessed with different WRF species for different treatment periods.

| Treatment Period (Days) | Fungal Species | IVDMD (g/100 g) | IVGP (mL/g OM) | | | $CH_4$ (% IVGP) | AFB1, µg kg$^{-1}$ |
|---|---|---|---|---|---|---|---|
| | | | Total | $CO_2$ | $CH_4$ | | |
| 0 | *A. bisporus* | 46.0 [a] | 156.3 [a] | 130.5 [a] | 22.8 [ab] | 14.6 [ef] | <5.0 |
| | *P. djamor* | 46.8 [a] | 159.0 [a] | 133.7 [a] | 22.4 [a] | 14.1 [ef] | <5.0 |
| | *C. indica* | 46.9 [a] | 159.0 [a] | 132.2 [a] | 22.6 [a] | 14.2 [ef] | <5.0 |
| | *P. ostreatus* | 46.1 [a] | 155.3 [a] | 130.2 [a] | 22.9 [ab] | 14.7 [f] | <5.0 |
| 21 | *A. bisporus* | 53.8 [c] | 176.0 [e] | 148.2 [e] | 24.4 [bcdef] | 13.4 [bcd] | <5.0 |
| | *P. djamor* | 53.4 [c] | 190.0 [d] | 165.1 [d] | 23.8 [cdef] | 12.6 [de] | <5.0 |
| | *C. indica* | 64.6 [a] | 203.7 [c] | 177.2 [c] | 23.7 [def] | 11.6 [ef] | <5.0 |
| | *P. ostreatus* | 55.7 [bc] | 193.3 [d] | 165.9 [d] | 24.6 [bcde] | 12.7 [cde] | <5.0 |
| 56 | *A. bisporus* | 59.2 [c] | 188.3 [c] | 160.2 [c] | 25.2 [cdef] | 13.1 [cd] | <5.0 |
| | *P. djamor* | 57.9 [c] | 208.0 [de] | 180.5 [e] | 25.7 [def] | 12.4 [bc] | <5.0 |
| | *C. indica* | 65.1 [d] | 237.3 [f] | 205.9 [f] | 25.9 [ef] | 10.9 [a] | <5.0 |
| | *P. ostreatus* | 60.5 [c] | 215.7 [e] | 185.8 [e] | 26.8 [f] | 12.4 [bc] | <5.0 |
| SEM | | 1.98 | 1.82 | 1.90 | 0.40 | 0.25 | |
| Overall mean of WRF species | | | | | | | |
| *A. bisporus* | | 53.2 [ab] | 173.6 [a] | 148.2 [a] | 24.1 [b] | 13.9 [c] | <5.0 |
| *P. djamor* | | 51.3 [a] | 185.7 [b] | 158.3 [b] | 24.0 [b] | 13.0 [b] | <5.0 |
| *C. indica* | | 58.3 [c] | 200.0 [c] | 174.2 [d] | 24.0 [b] | 12.2 [a] | <5.0 |
| *P. ostreatus* | | 54.2 [ab] | 188.1 [b] | 161.1 [c] | 24.8 [a] | 13.2 [b] | <5.0 |
| SEM | | 1.05 | 0.23 | 0.35 | 0.14 | 0.21 | |
| Overall mean treatment period | | | | | | | |
| 0 days | | 45.0 [a] | 157.4 [a] | 132.1 [a] | 22.7 [a] | 14.4 [c] | <5.0 |
| 21 days | | 56.6 [b] | 190.8 [b] | 164.3 [b] | 24.1 [b] | 12.7 [b] | <5.0 |
| 56 days | | 61.0 [c] | 212.3 [c] | 186.4 [c] | 25.9 [c] | 12.3 [a] | <5.0 |
| SEM | | 1.29 | 0.91 | 0.88 | 0.20 | 0.12 | |
| Significance | | | | | | | |
| WRF species | | * | *** | *** | NS | *** | |
| Treatment period | | *** | *** | *** | *** | *** | |
| WRF species × treatment period | | * | ** | ** | NS | * | |

Values with different superscripts in columns for treatment periods × fungal species, overall means of fungal species or overall mean of treatment periods are significantly ($p < 0.05$) different. OM, organic matter; $CH_4$, methane; $CO_2$, carbon dioxide; AFB1, aflatoxin B1. NS, non-significant ($p > 0.05$); *, $p < 0.05$; **, $p < 0.01$; ***, $p < 0.001$.

Whereas $CH_4$ production (% of IVGP) decreased ($p < 0.05$). Further comparison of the interaction data revealed that the greatest ($p < 0.05$) increase in IVDMD and IVGP and greatest ($p < 0.05$) decrease in $CH_4$ (% of IVGP) were achieved through the bioprocessing of SCB for 56 days with *C. indica*. The lowest increase in IVGP and IVDMD and the lowest decrease in $CH_4$ (% of IVGP) after 21 and 56 days of treatment were observed for *A. bisporus*. Notably, SCB pretreated with *C. indica* for 21 days had greater ($p < 0.05$) IVGP (203.7 vs. 188.3 mL/g OM) and IVDMD (64.6 vs. 59.2%) values than those recorded for *A. bisporus* after 56 days. On the other hand, the decrease in $CH_4$ (% of IVGP) after 21 days of SCB treatment with *C. indica* was lower than the value recorded for *A. bisporus* after 56 days.

The overall mean of fungal species revealed the greatest ($p < 0.05$) IVDMD (58.3%) and IVGP (200 mL/g OM) and lowest ($p < 0.05$) $CH_4$ production (12.2% of IVGP) for SCB treated with *C. indica*. With the increase in treatment period from 0 to 56 days, the mean values of IVDMD (45.0 to 61.0%), IVGP (157.4 to 212.3 mL/g OM) and $CH_4$ production (22.7 to 25.9 mL/g OM) increased, whereas the proportion of $CH_4$ in total IVGP decreased (14.4 to 12.3%; $p < 0.05$) with the increase in the treatment period from 0 to 56 days (Table 3). The content of AFB1 after 0, 21 and 56 days treated with SCB was less than 5 µg kg$^{-1}$ for all WRF species (Table 3).

Figure 1 shows the rate and extent of IVGP (mL/g OM) during 72 h of incubation in buffered-rumen fluid of SCB bioprocessed with *A. bisporus* (Panel A), *P. ostreatus* (Panel B), *C. indica* (Panel C) and *P. djamor* (Panel D) for 0, 21 and 56 days. Irrespective of the fungal species, the rate and extent of IVGP increased with the pre-treatment of SCB for 21 and 56 days.

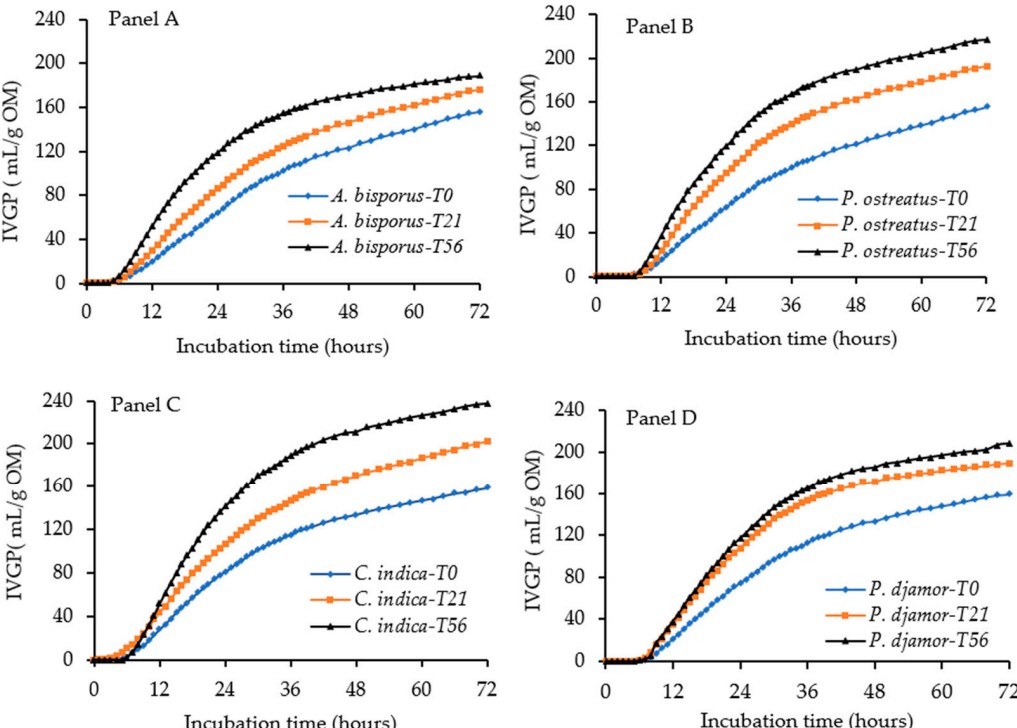

**Figure 1.** In vitro total gas production (IVGP; mL/g organic matter (OM)) for 72 h of incubation in buffered-rumen fluid of sugarcane bagasse treated with Panel (**A**) *Agaricus bisporus* (*A. bisporus*), Panel (**B**) *Pleurotus ostreatus* (*P. ostreatus*), Panel (**C**) *Calocybe indica* (*C. indica*) and Panel (**D**) *Pleurotus djamor* (*P. djamor*) for 0 (control; T0), 21 (T21) and 56 (T56) days.

Figure 2 clearly shows that the degradation of lignin during the bioprocessing of SCB was strongly associated with the increase in IVDMD ($R^2 = 0.72$; Panel A) and IVGP ($R^2 = 0.93$; Panel B). Moreover, the decrease in lignin-to-cellulose ratio (LCR) during the bioprocessing of SCB was strongly associated with the increase in IVDMD ($R^2 = 0.71$; Figure 3A) and IVGP ($R^2 = 0.95$; Figure 3B).

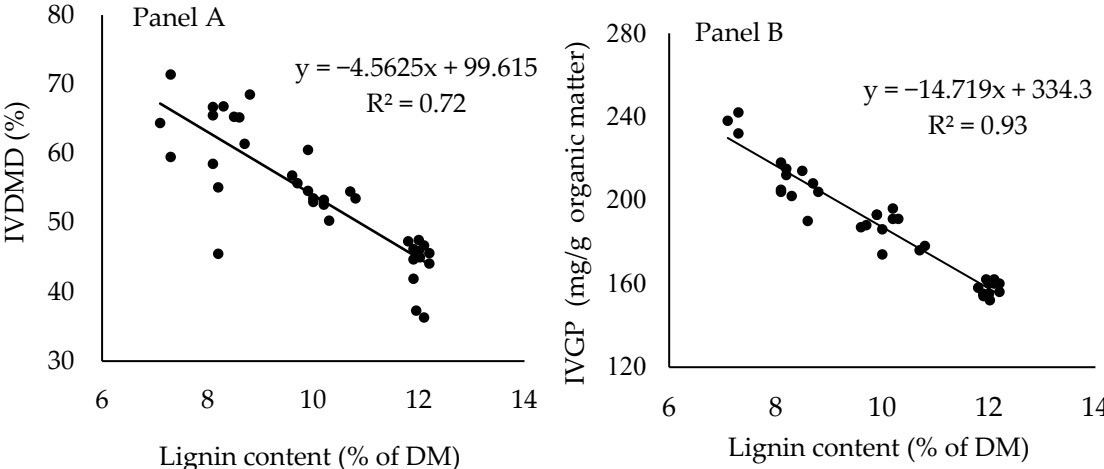

**Figure 2.** Relationship between lignin content and in vitro dry matter digestibility (IVDMD; Panel (**A**)) and in vitro total gas production (IVGP mL/g organic matter (OM); Panel (**B**)) of sugarcane bagasse pretreated with *Agaricus bisporus*, *Pleurotus ostreatus*, *Calocybe indica* and *Pleurotus djamor* for 0, 21 and 56 days.

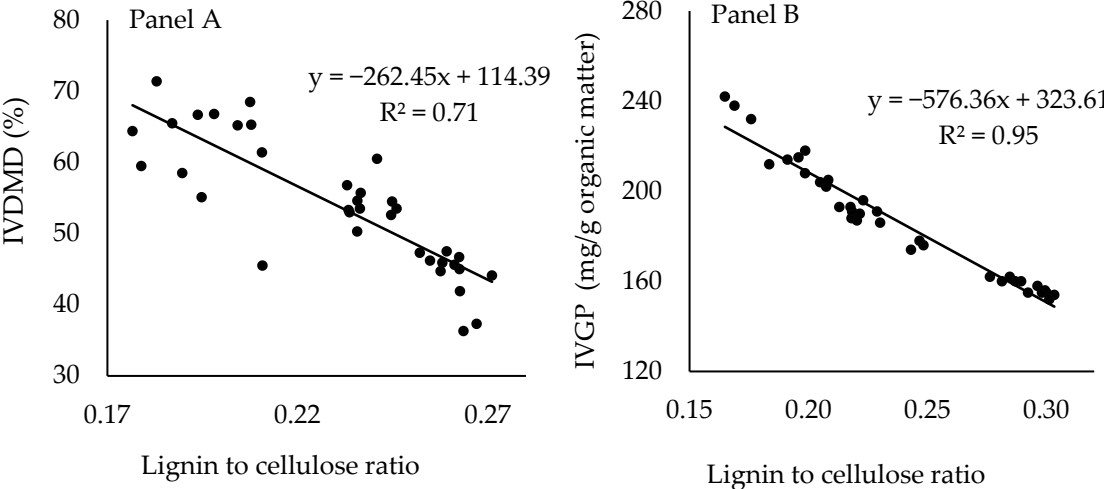

**Figure 3.** Relationship between lignin-to-cellulose ratio and in vitro dry matter digestibility (IVDMD; Panel (**A**)) and in vitro total gas production (IVGP; Panel (**B**)) of sugarcane bagasse pretreated with *Agaricus bisporus*, *Pleurotus ostreatus*, *Calocybe indica* and *Pleurotus djamor* for 0, 21 and 56 days.

## 4. Discussion

As in nature, during the mycelial colonization phase, WRF preferentially produces a complex mixture of oxidative ligninolytic enzymes and small radicals [30,31] to degrade lignin and increase the availability of polysaccharides for fruit body production. The fungal treatment is stopped before fruit body production, ideally at the stage when most of the lignin is degraded, and the maximum accessibility of cellulose is achieved for microbial fermentation in the rumen [13,14]. Lignin degradation during fungal treatment is the primary index for the improvement of the ruminal degradability of the recalcitrant structures of LCB. Nevertheless, a variable fraction of hemicellulose and cellulose is also degraded by WRF as an energy and growth substrate for mycelial colonization. Recent research has shown that the efficiency of fungal treatment of LCB not only depends on the delignification and the consequent increase in ruminal degradability but also on the availability of polysaccharides as an energy source for microbial fermentation [11,13,15]. When more polysaccharides are degraded, then the delignification process may not improve the fermentable energy supply for ruminant nutrition [20]. Therefore, the extent of lignin

and polysaccharides degradation, as reflected by the changes in LCR, is a major determinant for the availability of fermentable energy and the overall efficiency of the fungal treatment of LCB [11,14].

The losses of structural components during the treatment of SCB for 21 and 56 days in current study could be related to the natural ability of WRF to mineralize fractions of lignin, hemicellulose and cellulose into $CO_2$ and $H_2O$ [13,32]. Our results are consistent with the findings of previous studies, which were conducted on the other substrates [11,17,19]. Overall, lignin degradation in the present study ranged from 18.1 to 49.3%, with the greatest value being observed for *C. indica* after 56 days, followed by *P. ostreatus*, *P. djamor* and *A. bisporus*. On the other hand, cellulose degradation ranged from 10.0 to 25.6%, with the greatest values being observed for *C. indica* followed by *P. ostreatus*, *A. bisporus* and *P. djamor* (Table 2). Overall, cellulose degradation was lower than the lignin degradation for all WRF species tested in this study. As discussed earlier, the balance of polysaccharides in the treated biomass is very important, and as such, maximum lignin degradation with minimal cellulose degradation in the residual biomass is desirable for ruminant nutrition [11,33]. In the current study, the lowest degradation of cellulose was recorded for *P. djamor* compared to other WRF species. However, the comparison of the relative degradation of lignin and cellulose as expressed by LCR (calculated from Table 1) revealed that *C. indica* had lowest LCR (0.18) as compared to other WRF species (LCR ranged from 0.20 to 0.28). Next, in terms of the extent and selectivity for lignin degradation, *C. indica* also degraded lignin more quickly. In the present study, *C. indica* degraded more lignin after 21 days than those degraded by other WRF species after 56 days. Speed, selectivity and extensive delignification are the perquisites for fungal treatments of greater efficiency [34,35]. Nayan et al. [25] reported that *Ceriporiopsis subvermispora* strains degraded lignin more rapidly and selectively during the treatment of wheat straw, resulting in the greater improvement in cellulose availability and IVGP as compared to other fungal species. Datta et al. [36] reported the faster mycelial colonization of *C. indica* on SCB, as reflected by the earliest formation of pinheads (end of vegetative growth; at day 24), as compared to the other substrates (ranging from day 28 to 40). These findings highlight the potential of *C. indica* for the greater and more robust synthesis of mycelia when grown on SCB. The rapid colonization of mycelium in the substrate is a prerequisite for an effective fungal treatment. Fungi degrade lignin by releasing extracellular enzymes that have very limited ability to diffuse through the dense lignocellulosic structure. Therefore, the rate and extent of mycelial colonization and the subsequent penetration deep into the cell wall matrix are closely associated with the rate and extent of lignin degradation and the efficiency of the fungal treatment [25,37,38]. Moreover, greater biological efficiency for mushroom production has been reported for *C. indica* when grown on SCB [36], which indirectly support our findings. The greater extent of lignin degradation and lower LCR demonstrated that *C. indica* is a new potential candidate for the bioprocessing of SCB.

Notably, all WRF species consistently degraded lignin in SCB during the 56-day treatment period. However, the greater degradation of lignin was recorded during the initial phase (0–21 days) as compared to the later phase (22–56 days). In agreement with our findings, Mao et al. [39] and Van Kuijk et al. [35] reported greater lignin degradation during the initial phase (0–28 days) of the treatment of wheat straw with *C. subvermispora* as compared to the later phase (28–56 days). The substantially higher losses of hemicellulose were also observed during the initial phase (25.5%) as compared to the final phase (7.4%), which demonstrates that the fungal mycelia consumed the easily fermentable carbohydrates for its initial growth. The initial robust growth of fungal mycelia is a prerequisite to ensure desirable fermentation in the substrate and to inhibit the growth of contamination-causing microbes [39]. Such patterns of hemicellulose consumption by WRF have also been reported in recent studies [11,40].

The marked increase in the CP content was observed after 21 and 56 days of treatment of SCB for all WRF species. During fungal treatment, the CP content of the substrate (indirectly) increases due to the degradation of cell wall components into $CO_2$ and $H_2O$,

which causes significant losses of OM, while N remains intact, increasing the concentration of CP in the residual mass. WRF assimilate nitrogenous compounds from the degraded substrate [41], which reduces losses and contributes to the increase in the CP content of the treated substrate. The greater increase in the CP contents of SCB pretreated with *C. indica* (104.1%) can be related to the greater degradation of lignin and structural polysaccharides by *C. indica* as compared to the other WRF species. In addition to the indirect proportional increase, higher increases in CP contents by *C. indica* may also be attributed to the greater mycelial growth and laccase activity of this fungus as compared to other fungal species [42,43], which contributes to a higher CP content in the residual mass of SCB.

In the current study, all tested WRF species exhibited markedly increased IVDMD and IVGP after 21 and 56 days of treatment of SCB under SSF. The increase in IVDMD and IVGP with the fungal treatment could be related to the decrease in lignin and fiber contents and to the faster and more extensive degradation of the treated biomass. During fungal treatment, the soluble, easily degradable, non-fiber carbohydrate fractions increased, whereas the recalcitrant lignin and least degradable fiber fractions decreased, resulting in the greater degradability of treated biomass [3,33]. Moreover, fungal treatment changes the physical structure of the biomass in such a way that it becomes more brittle, causing a more rapid reduction in particle size and ruminal degradability. Finally, the improvement in the CP content of the treated biomass is also expected to increase the IVDMD and IVGP of treated biomass [20]. In agreement with our findings, earlier studies have reported similar increases in IVDMD and IVGP for different LCB after fungal treatment [11,16,35].

Irrespective of WRF species and treatment periods, the increase in IVDMD and IVGP was strongly associated with the degradation of lignin during the fungal treatment of SCB (Figure 2). Notably, the increase in IVGP was strongly related to the decrease in LCR during the treatment of SCB, irrespective of fungal species and treatment periods (Figure 3). It may be noted that IVDMD represents the solubility of SCB, whereas IVGP represents the ruminal fermentation of OM in SCB, which could explain the stronger relationship between LCR and IVGP. It is possible that some of the fungal-degraded products, such as the small monomers of lignin, may not be fermentable in the rumen. Therefore, IVGP truly represents the fermentation of the substrate in the rumen and the supply of fermentable organic matter to ruminant animals [14]. This further demonstrates that in addition to lignin degradation, other factors, such as the availability of cell wall polysaccharides, also contribute to the increase in fermentable OM supply for the animals. LCR and IVGP appear to be the most important indices for the efficiency of fungal treatment.

Of the fungal species, the greatest improvements in IVDMD and IVGP were through *C. indica*, which may be related to the greatest lignin degradation and the greatest decrease in LCR caused by the same fungus. Such a relationship between ruminal degradation characteristics and lignin contents for WRF-treated wheat straw, rice straw and corncobs has been demonstrated in previous studies [11,33,37]. Moreover, a persistent increase in IVGP from 21 to 56 days of treatment was observed with *C. indica* as compared to the other fungal species, which may be due to the more selective and persistent lignin degradation by this species, as discussed previously. On the other hand, after 21-day treatment, *P. djamor* was the most efficient in terms of increase in IVGP per unit of DM loss (for each gram of DM loss, values were 58 mL and 38.9 mL for day 21 and day 56, respectively), as compared to the other fungal species (14.3 to 26.3 mL/g DM loss) (calculated in Tables 1 and 2). The greater increase in IVGP with smaller losses of lignin after 21-days treatment of SCB with *P. djamor* is a question for further research. Details on the production of extracellular enzymes and changes in lignin–cellulose complexes and the physical structures of SCB and their influences on rumen microbes and ecosystems could explain the underlying mechanisms involved.

A significant outcome of this study is the decrease in the $CH_4$ fraction of total IVGP after the treatment of SCB by all fungi, and this reduction in $CH_4$ was more pronounced for *C. indica*. The obvious reason for the reduction in $CH_4$ production was the decrease in cellulose and hemicellulose contents and the greater rate and extent of the ruminal

fermentation of the treated SCB in this study. Sun et al. [44], in a comprehensive review, demonstrated that the fermentation capability of methanogenic microbes in the rumen is higher for β-1,4-linked structural polysaccharides when compared to non-structural polysaccharides. In agreement with this study, Huyen et al. [45] also reported a decrease of 11.4% in $CH_4$ production, after 24 h of in vitro incubation, in SCB treated with *P. eryngii* for 4 weeks.

Sugarcane bagasse naturally contains a very minute amount of AFB1 (1.55 μg kg$^{-1}$ DM) [45] which was not detected in the screening test in this study. Moreover, some WRF degrades the AFB1 by a range of 40 to 94.7% [46,47]. Therefore, in the present study, WRF were less likely to have detectable levels of AFB1 in treated SCB regardless of the type of fungi, and it is inferred that SCB treated with all fungi used in this study was safe to feed to ruminant animals.

A significant interaction between the WRF species and treatment period for an improvement in the nutritional value and ruminal fermentation characteristics of SCB demonstrates the scope for the optimization of this biotechnology to achieve a more desirable nutritional upgradation in less time. For example, *C. indica* can be used to fast-track the delignification process during fungal treatment. In the present study, the 21-day treatment with *C. indica* was more or comparably effective than the 56-day treatment with the other fungal species in terms of the delignification and improvement of the nutritional value and ruminal fermentability of the treated SCB. Further studies are suggested with more treatment periods (like obtaining data on a weekly basis) and involving other factors like increasing the amount of inoculum, optimizing the chop size of the substrate and supplementing essential nutrients for a more precise optimization for the treatment period and the nutritional upgradation of SCB using WRF. The extraction and application of bacterial and fungal ligninolytic enzymes can also be used to improve the degradability of LCB [48,49].

Despite the promising results of fungal treatment on lignin-rich biomass for ruminant feeding at the laboratory level in this and previous studies [11,16,17], there are questions regarding the practical feasibility of this technique at a farm or commercial level. To replicate the technique at a larger scale, it is important to consider certain challenging aspects, which include the large-scale disinfection of the substrate and inoculation process, the shortening of the treatment period and the drying of mass quantities of fungal treated biomass to stop the fermentation process and for storage. In one of our recent studies [3], we addressed the issue of mass-scales and the feasible disinfection of the substrate for fungal treatment. In another study on the feeding of fungal-treated biomass to dairy cows [50], which is in the process of publication, we demonstrated that DM intake and milk production in cows was increased when they were fed WRF-treated biomass.

## 5. Conclusions

This study revealed that selective lignin-degrading WRF species, namely *A. bisporus*, *P. ostreatus*, *C. indica* and *P. djamor*, can be used to degrade lignin and improve the nutritional value and digestibility of SCB for ruminants. Of the fungi, *C. indica* not only caused the highest degradation of lignin but also resulted in the smallest lignin-to-cellulose ratio in the treated SCB—therefore sparing more cellulose for ruminal microbial fermentation, which is evident by the highest in vitro digestibility and gas production of SCB treated with *C. indica*. Except for *A. bisporus*, all fungi reduced in vitro $CH_4$ production with the highest reduction caused by *C. indica*. The increase in gas production was strongly associated with lignin degradation ($R^2 = 0.72$), and the decrease in the lignin-to-cellulose ratio ($R^2 = 0.95$) during the bioprocessing of SCB. *C. indica* presents great prospects for the rapid, selective and more extensive degradation of lignin and, as such, for the improvement in nutritional value and digestibility of SCB for ruminant nutrition.



**Author Contributions:** Conceptualization, N.A.K., Z.T. and S.T.; methodology, M.K., A.S. (Abubakar Sufyan) and A.S. (Ashmal Saeed); software, N.A.K.; formal analysis, N.A.K., A.S. and M.N.; Investigation, M.K., A.S. (Abubakar Sufyan) and A.S. (Ashmal Saeed); writing—original draft preparation, M.K. and M.N.; writing—review and editing N.A.K., A.S. (Abubakar Sufyan), L.S., S.W. and Y.L.; visualization, Z.T. and S.T.; supervision, N.A.K.; project administration, N.A.K., Z.T. and S.T.; Financial support, Z.T., Y.L. and S.T. All authors have read and agreed to the published version of the manuscript.

**Funding:** The experimental trial was financially supported by Chinese Academy of Sciences President's International Fellowship for Postdoctoral Studies (2021BP0045), the Strategic Priority Research Program of the Chinese Academy of Sciences (grant No. XDA28020400), and the Rural Revitalization Project of Chinese Academy of Sciences (KFJ-XCZX-202303). The financial support from the Higher Education Commission of Pakistan (grant IDs 2AV3-017 and IRSIP-44-Agri-01) to carry out laboratory work at Southern Illinois University (SIU), Carbondale, USA, is highly appreciated.

**Institutional Review Board Statement:** Not applicable.

**Informed Consent Statement:** Not applicable.

**Data Availability Statement:** Data available on reasonable demand from the corresponding author.

**Acknowledgments:** The technical guidance and support from Amer AbuGhazaleh and Mohamed Galal Embaby (PhD scholar) at SIU is acknowledged and greatly appreciated.

**Conflicts of Interest:** The authors declare no conflicts of interest.

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
