# Peer review of "Biotechnological Processing of Sugarcane Bagasse through Solid-State Fermentation with White Rot Fungi into Nutritionally Rich and Digestible Ruminant Feed"

_fermentation, doi:10.3390/fermentation10040181_

Round 1

Reviewer 1 Report

Comments and Suggestions for Authors

The paper titled " Biotechnological Processing of Sugarcane Bagasse Through Solid State Fermentation with White Rot Fungi into Nutritionally Rich and Digestible Ruminant Feed” examines the feasibility of utilizing selective lignin-degrading White Rot Fungi (WRF) species to enhance the nutritional value and digestibility of Sugarcane Bagasse (SCB) for potential use in ruminant feeding. In this study, four fungi species were used: A. bisporus, P. ostreatus, C. indica, and P. djamor.

The results obtained are both intriguing and practically significant. The authors discovered that treating SCB with selective lignin-degrading WRF can enhance its nutritional value and digestibility. Notably, C. indica shows promising potential for rapid, selective, and more extensive lignin degradation, thereby improving the nutritional quality and digestibility of SCB for ruminant nutrition. The manuscript is well-written.

The fungi species (A. bisporus, P. ostreatus, C. indica, and P. djamor) are commonly used in culinary practices due to their popularity, taste, texture, and nutritional content. The criteria guiding the selection of these species for the study remain unspecified. I recommend a brief addition to the introduction section, elucidating the criteria for choosing these fungi species and pointing whether similar studies have been conducted previously. This could help to determine if the research is pioneering. Furthermore, it would be valuable to discuss the practical application of the findings. Were there considerations for animal feed testing? Is it know how does the application of Solid-State Fermentation (SSF) impact the palatability of the feed?

Author Response

Reviewer 1

The paper titled " Biotechnological Processing of Sugarcane Bagasse Through Solid State Fermentation with White Rot Fungi into Nutritionally Rich and Digestible Ruminant Feed” examines the feasibility of utilizing selective lignin-degrading White Rot Fungi (WRF) species to enhance the nutritional value and digestibility of Sugarcane Bagasse (SCB) for potential use in ruminant feeding. In this study, four fungi species were used: A. bisporus, P. ostreatus, C. indica, and P. djamor. The results obtained are both intriguing and practically significant. The authors discovered that treating SCB with selective lignin-degrading WRF can enhance its nutritional value and digestibility. Notably, C. indica shows promising potential for rapid, selective, and more extensive lignin degradation, thereby improving the nutritional quality and digestibility of SCB for ruminant nutrition. The manuscript is well-written. C indica is also native to subcontinent.

AU: Thanks very much for a very comprehensive reviews, and for your kind and positive remarks about the research study and manuscript.

The fungi species (A. bisporus, P. ostreatus, C. indica, and P. djamor) are commonly used in culinary practices due to their popularity, taste, texture, and nutritional content. The criteria guiding the selection of these species for the study remain unspecified. I recommend a brief addition to the introduction section, elucidating the criteria for choosing these fungi species and pointing whether similar studies have been conducted previously. This could help to determine if the research is pioneering.

AU: Suggestion well received. Yes, mostly edible white rot fungi species are used for bioprocessing of lignocellulose biomass for ruminant nutrition. Information about the selection of fungal species for this study added in the introduction section.

Furthermore, it would be valuable to discuss the practical application of the findings. Were there considerations for animal feed testing? Is it know how does the application of Solid-State Fermentation (SSF) impact the palatability of the feed?

AU: Suggestion well received. Information about animal feeding trials, palatability of the treated biomass, and practical application of the fungal treated biomass in ruminant nutrition are provided in the discussion section of the revised manuscript.

Reviewer 2 Report

Comments and Suggestions for Authors

 Despite the copious literature available on the topic, the manuscript Biotechnological Processing of Sugarcane Bagasse Through 2 Solid State Fermentation with White Rot Fungi into Nutrition-3 ally Rich and Digestible Ruminant Feed represents an advance in knowledge as the mushroom species used in this trial have never been tested before for the degradation of SCB.

The manuscript is overall well-written although some aspects examined in detail below need to be revised

Line 37: explain the acronym

Line 53: more recent data can be found in the following paper

Ungureanu, Nicoleta, Valentin VlăduÈ›, and Sorin-Ștefan BiriÈ™. 2022. "Sustainable Valorization of Waste and By-Products from Sugarcane Processing" Sustainability 14, no. 17: 11089. https://doi.org/10.3390/su141711089

Line 165: Add country to equipment specifications

Line 191: please describe more thoroughly the rumen sampling. The quantity of sampling-  was it taken from different regions of the rumen - was it mixed from the four animals - was homogenized to detach fungi from feed particles.

Please add the diet administered to the cannulated animals and its chemical analysis. Diet could influence the degradation rate during gas production.  

Line 202 and 209: see 165

Table 1.  Why A bisporus is in bold?.

The superscript letter a-b-c-.. should be written starting from the lowest to the highest

Table 2: report units of measurement

Figure 2 caption: It is not Panel 2 but Panel B

As a general observation, the discussion could be deepened.

Line 378- 392 and 452-453: avoid repetition of results

Line 436-439: the paragraph is not necessary because the SCB was sterilized as you said

Line 466-478: Please avoid the repletion and add references for discussions

Line 489: leave space before as this is a new concept

Revise  references  carefully according to journal requirements  the majority are not formatted as expected

Author Response

Dear Ms. Marciana Barbulescu
Assistant Editor

We are very thankful to you for timely processing this manuscript and to the reviewers for a comprehensive review and very valuable comments. Overall, these comments allowed us to improve the quality of this manuscript (ID: fermentation-2886573) considerably. We have carefully addressed the comments of the reviewers, and provide explanation to few comments. We are pleased to resubmit a revised manuscript to be considered for publication in Fermentation.

Reviewer 2

Despite the copious literature available on the topic, the manuscript Biotechnological Processing of Sugarcane Bagasse Through 2 Solid State Fermentation with White Rot Fungi into Nutrition-3 ally Rich and Digestible Ruminant Feed represents an advance in knowledge as the mushroom species used in this trial have never been tested before for the degradation of SCB.

AU: Thanks very much for a very comprehensive reviews, and for your kind and positive remarks about the research study and manuscript.

The manuscript is overall well-written although some aspects examined in detail below need to be revised

AU: Thanks very much. We have carefully addressed all the comments.

Line 37: explain the acronym

AU: IVGP and WRF, are both explained/defined in line number 29 and 34, respectively.

Line 53: more recent data can be found in the following paper

Ungureanu, Nicoleta, Valentin VlăduÈ›, and Sorin-Ștefan BiriÈ™. 2022. "Sustainable Valorization of Waste and By-Products from Sugarcane Processing" Sustainability 14, no. 17: 11089. https://doi.org/10.3390/su141711089

AU: Suggestion well received, most recent data from FAOSTAT (2023) and Ungureanu et al (2022) have been included.

Line 165: Add country to equipment specifications

AU: Done

Line 191: please describe more thoroughly the rumen sampling. The quantity of sampling-  was it taken from different regions of the rumen - was it mixed from the four animals - was homogenized to detach fungi from feed particles.

AU: Required details are provided in the revised manuscript

Please add the diet administered to the cannulated animals and its chemical analysis. Diet could influence the degradation rate during gas production.

AU: Suggestion well received, the ingredient and chemical composition of diet fed to the cannulated animals are included in the revised manuscript.

Line 202 and 209: see 165

AU: Details of equipment used are included.

Table 1.  Why A bisporus is in bold?.

AU: That was a formatting error. It is corrected.

The superscript letter a-b-c-.. should be written starting from the lowest to the highest

AU: The superscripts were rearranged as suggested. However, suggestion was applied on all tables to avoid any confusion to the readers.

Table 2: report units of measurement

AU: Unit of measurement included

Figure 2 caption: It is not Panel 2 but Panel B

AU: Corrected

As a general observation, the discussion could be deepened.

AU: Agreed, where possible we have deepened the discussion in the revised manuscript.

Line 378- 392 and 452-453: avoid repetition of results

AU: Results deleted.

Line 436-439: the paragraph is not necessary because the SCB was sterilized as you said

AU: Unnecessary phrase is removed as suggested.

Line 466-478: Please avoid the repletion and add references for discussions

AU: Duplication removed and references are added.

Line 489: leave space before as this is a new concept

AU: Space incorporated as suggested.

Revise references carefully according to journal requirements the majority are not formatted as expected

AU: References revised according to the journal format.

Round 2

Reviewer 2 Report

Comments and Suggestions for Authors

Thank you to the authors, the manuscript has been successfully improved